# Association of Epicardial Adipose Tissue Adipocytes Hypertrophy with Biomarkers of Low-Grade Inflammation and Extracellular Matrix Remodeling in Patients with Coronary Artery Disease

**DOI:** 10.3390/biomedicines11020241

**Published:** 2023-01-17

**Authors:** Irina V. Kologrivova, Natalia V. Naryzhnaya, Olga A. Koshelskaya, Tatiana E. Suslova, Elena S. Kravchenko, Olga A. Kharitonova, Vladimir V. Evtushenko, Alla A. Boshchenko

**Affiliations:** Cardiology Research Institute, Tomsk National Research Medical Center, Russian Academy of Sciences, 111A Kievskaya, 634012 Tomsk, Russia

**Keywords:** epicardial adipose tissue, adipocytes, secreted phospholipase A2, interleukin-1β, tumor necrosis factor, C-terminal cross-linking telopeptide of type I collagen

## Abstract

The aim of the study was to compare the morphological features of epicardial adipose tissue (EAT) adipocyte with the circulating inflammatory biomarkers and parameters of extracellular matrix remodeling in patients with coronary artery disease (CAD). We recruited 42 patients with CAD (m/f 28/14) who were scheduled for coronary artery bypass graft surgery (CABG). EAT adipocytes were obtained by the enzymatic method from intraoperative adipose tissue samples. Concentrations of secreted and lipoprotein-associated phospholipase A2 (sPLA2 and LpPLA2), TNF-α, IL-1β, IL-6, IL-10, high-sensitive C-reactive protein (hsCRP), metalloproteinase-9 (MMP-9), MMP-2, C-terminal cross-linking telopeptide of type I collagen (CTX-I), and tissue inhibitor of metalloproteinase 1 (TIMP-1) were measured in blood serum. Patients were divided into two groups: group 1—with mean EAT adipocytes’ size ≤ 87.32 μm; group 2—with mean EAT adipocytes’ size > 87.32 μm. Patients of group 2 had higher concentrations of triglycerides, hsCRP, TNF-α, and sPLA2 and a lower concentration of CTX-I. A multiple logistic regression model was created (R_N_^2^ = 0.43, *p* = 0.0013). Concentrations of TNF-α, sPLA2 and CTX-I appeared to be independent determinants of the EAT adipocyte hypertrophy. ROC analysis revealed the 78% accuracy, 71% sensitivity, and 85% specificity of the model, AUC = 0.82. According to our results, chronic low-grade inflammation and extracellular matrix remodeling are closely associated with the development of hypertrophy of EAT adipocytes, with serum concentrations of TNF-α, sPLA2 and CTX-I being the key predictors, describing the variability of epicardial adipocytes’ size.

## 1. Introduction

The deposition of ectopic fat depots (within the liver, pancreas, heart, kidneys, and skeletal muscles) represents one of the examples of adiposopathy, which is a recently coined term describing abnormal response of adipose tissue to positive caloric balance [1]. Epicardial adipose tissue (EAT), located between the myocardium and epicardium, predominantly in the atrioventricular and interventricular grooves and free wall of the right ventricle, has attracted the closest attention as one of the major players in the development of atherosclerosis [2]. No fascia separates it from the myocardium, giving it contiguity with the heart muscle. EAT is unique not only due to its close proximity to coronary arteries but also due to its transcriptome and proteome, which is different from that both in subcutaneous adipose tissue (SAT) and in other visceral fat depots [3]. However, the pathogenic factors that aid in determining the development of dysfunctional EAT in patients with CAD remain poorly studied.

Chronic low-grade inflammation represents one of the main driving forces leading to the malfunctioning of adipose tissue [4] and is regarded as a pathophysiological link between dysfunctional EAT and emergence of cardiovascular disorders [5,6]. The opposite is also true, as an increase in the adipocytes’ number and size creates local hypoxia, followed by the release of free fatty acids and inflammatory cytokines, which ultimately leads to an accumulation of macrophages [7]. An influx of immune cells to the stromal vascular fraction of adipose tissue polarized toward inflammatory phenotype induces the development of inflammatory changes and skews the properties of adipocytes [4]. Inflammatory pathways are involved in the expansion of adipose tissue and its extracellular matrix remodeling [8]. EAT has been proven to be more inflammatory than subcutaneous depots based on the evaluation of chemokines, cytokines and immune cells content [9,10]. Expression of the major inflammatory cytokine, tumor necrosis factor (TNF)-α, was enhanced in EAT compared to SAT [11]. The inflammatory environment is supported by various mediators, including phospholipase A2 (PLA2), which exists in lipoprotein-associated (LpPLA2) and secreted forms (sPLA2) and is produced in various tissues [12].

Despite the emerging data on the implication of EAT thickness and volume in systemic metabolism and cardiovascular risk, the morphology of EAT adipocytes remains rather poorly investigated. The total volume of EAT may increase either via the enlargement of adipocytes (hypertrophy) or via increase in the fat cells number (hyperplasia) [13]. The average size of adipocytes has been shown to reflect the systemic function of fat cells, with large, hypertrophic cells being less insulin sensitive and associated with metabolically compromised phenotype [14]. Not only the mean size of adipocytes but the proportion of large and small adipocytes appeared to matter in maintaining of the normal function of adipose tissue [15]. Both extra-large and extra-small fat cells demonstrated an inability to expand appropriately, which favored the development of metabolic impairments and insulin resistance [16]. 

A number of underlying mechanisms may be suspected, causing inadequate expansion during the excessive supply of energy, including impaired glucose tolerance, increased insulin resistance and unfavorable extracellular matrix (ECM) remodeling [15,17]. The ECM in adipose tissue, consisting of collagens (I, VII), fibronectin, elastin, glycosaminoglycan (GAG) and laminin, controls cell function and adipogenesis [18]. Its remodeling is a key event during adipose tissue enlargement; however, it received little attention so far, especially in EAT. High-caloric intake is associated with the modulation of ECM turnover, refueling inflammation, and creating a vicious cycle. 

We hypothesized that the morphological characteristics of EAT adipocytes are associated with systemic metabolism, markers of chronic low-grade inflammation and extracellular matrix remodeling. The aim of the present study was (1) to compare morphological features of EAT adipocyte with anthropometric parameters of obesity, the state of systemic metabolism, and circulating levels of inflammatory and ECM remodeling biomarkers in patients with CAD undergoing coronary artery bypass grafting (CABG) surgery and (2) to reveal potential factors, providing adipose–metabolic–inflammatory cross-talk.

## 2. Materials and Methods

### 2.1. Patients

The present pilot study enrolled 42 patients (28 men and 14 women) aged 53–72 y.o. with chronic stable CAD and manifest coronary atherosclerosis with indications for CABG. 

The study is conducted in accordance with the guidelines of the Declaration of Helsinki with amendments as of 2000 and “Rules of Clinical Practice in the Russian Federation” approved by the Order of the Ministry of Health of the Russian Federation in 19 June 2003 No. 266. The study’s protocol was approved by the Biomedical Ethics Committee of Cardiology Research Institute, Tomsk NRMC (protocol № 210 from 18 February 2021). All the individuals recruited to the study provided their informed consent.

All the patients received standard conventional therapy. Characteristics of patients are represented in Table 1. 

Exclusion criteria included cardiovascular complications in the preceding 6 months (acute coronary syndrome, transient ischemic attack, etc.); active, ongoing inflammatory diseases other than atherosclerosis; history of active, ongoing or recurrent infections; chronic kidney disease class above C3b; decompensated diabetes mellitus; cancer; hematological and autoimmune disorders; change in body weight of more than 3% in the previous 3 months; refusal to participate in the study.

Anthropometric measurements were performed to assess total obesity according to the level of body mass index (BMI) and abdominal obesity according to the size of the waist circumference, hip circumference, and waist-to-hip ratio (WHR).

All the patients underwent selective coronary angiography on a Artis one angiographic complex and Digitron-3NAC computer system (Siemens Shenzhen Magnetic Resonance Ltd., Shenzhen, China). The severity of atherosclerosis was estimated via calculation of Gensini Score [19] and the number of vessels with obstructive atherosclerotic disease. Multivessel CAD was defined as stenosis of at least 70% in at least two major coronary arteries or in one coronary artery together with 50% or greater stenosis of the left main trunk. 

### 2.2. Echocardiography

The parasternal long-axis view was examined. EAT thickness was measured on the free wall of the right ventricle in a still image at the end-diastole [20]. The measurements were performed at the point of perpendicular orientation of the ultrasound beam to the free wall of the right ventricle, using the aortic annulus as an anatomic landmark. The EAT thickness was calculated as an average value from echocardiographic views in 3 consecutive cardiac cycles. 

### 2.3. Adipose Tissue Sampling and Processing

Tissue samples of EAT and SAT were obtained in the course of CABG in the amount of 0.2–1 g as described before [21]. Briefly, the samples were placed in M199 medium preheated to 37 °C in advance and delivered to the laboratory within 15 min. Adipocytes were isolated enzymatically in sterile conditions (laminar box BAVp-01- “Laminar-s” −1.5, ZAO “Laminar systems”, Miass, Russia) [22]. Adipose tissue was minced and incubated during 35–40 min at 37 °C and constant mild shaking (10 rpm) in 5 mL of collagenase type I solution (PanEco, Moscow, Russia) 1 mg/mL in Krebs–Ringer buffer (2 mM D-glucose, 135 mM NaCl, 2.2 mM CaCl_2_·2H_2_O, 1.25 mM MgSO_4_·7H_2_O, 0.45 mM KH_2_PO_4_, 2.17 mM Na_2_HPO_4_, 25 mM HEPES, 3.5% BSA, 0.2 mM adenosine). Krebs–Ringer buffer (37 °C) was added to the digested tissue to neutralize collagenase in 1:1 ratio. The suspension of cells was filtered through the nylon mesh (Falcon™Cell strainer, Glendale, AZ, USA, 100 μm) and rinsed three times with warm Krebs–Ringer buffer. The numbers and size of the obtained adipocytes were calculated in a Goryaev cell chamber by light microscopy (Axio Observer.Z1, Carl Zeiss Surgical GmbH, Oberkochen, Germany). The representative images of EAT adipocytes are displayed in Figure 1. Cells were stained with Hoechst 33,342 (5 μg/mL, stains nucleus of viable cells) and propidium iodide (10 μg/mL, Sigma-Aldrich, St. Louis, MO, USA, stains nucleus of dead cells) to distinguish viable cells from dead cells [23]. Samples with viability lower than 95% were excluded from the study. The remaining samples did not differ significantly in the percentage of viable cells.

### 2.4. Blood Sampling and Processing

Blood samples were obtained at least one day prior to the scheduled CABG after overnight fasting. Samples were centrifuged for 10 min at 1000× *g*. Serum was aliquoted and stored at −40 °C until the final analysis. 

### 2.5. Enzyme-Linked Immunosorbent Assay (ELISA)

Concentrations of insulin, C-peptide (both—AccuBind kits, Diagnostic System Laboratories, Lake Forest, CA, USA), hsCRP (Biomerica, Irvine, CA, USA), TNF-α, IL-1β, IL-6, IL-10 (all cytokine kits—VECTOR-BEST, Novosibirsk, Russia), LpPLA2 (RayBiotech, Norcross, GA, USA) and sPLA2 (Cayman Chemical, Ann Arbor, MI, USA), C-terminal cross-linking telopeptide of type I collagen (CTX-I) (Immunodiagnostic Systems, Gaithersburg, MD, USA), matrix metalloproteinase type 2 (MMP2) (R&D, Minneapolis, MN, USA), MMP9 (Affymetrix, Santa Clara, CA, USA), tissue inhibitor of matrix metalloproteinases type 1 (TIMP-1) (Affymetrix, Santa Clara, CA, USA), apolipoprotein B, and apolipoprotein A1 (both—AssayPro, St. Charles, MO, USA) were measured in serum by ELISA on an instrument Infinite F500 (Tecan, Männedorf, Switzerland). 

### 2.6. Biochemical Assays

The level of glucose was detected by hexokinase assay (EKF diagnostic, Leipzig, Germany). Enzyme colorimetric method was used to estimate the serum concentration of total cholesterol, triacylglycerol, high-density lipoprotein (HDL) cholesterol (Diakon, Pushchino, Russia). Concentration of low-density lipoprotein (LDL) cholesterol was calculated using the following formula: [LDL] = [Total cholesterol] − [Triglycerides (TG)] − [HDL].

### 2.7. Statistical Analysis

Statistical analysis was performed using Statistica 10.0 software (StatSoft Inc., Tulsa, OK, USA). The normality of the distribution of sample data was verified by the Shapiro–Wilk test. Results were expressed as median and interquartile interval (Q1; Q3) when distribution was different from normal. Categorical data were represented as absolute numbers (*n*) and frequencies (%). The Mann–Whitney rank sum test was used to assess statistically significant differences in independent groups for quantitative parameters, and Pearson’s chi-square test was used for categorical parameters. The Spearman correlation coefficient (r_s_) was used to study correlations between variables. Multiple logistics regression was used to estimate the prognostic significance of variables, predicting the degree of EAT adipocytes hypertrophy. All the statistical hypotheses were checked when the critical level of significance *p* = 0.05.

## 3. Results

### 3.1. Morphometric and Biochemical Parameters

Medians of the EAT adipocytes’ mean size in patients with coronary atherosclerosis constituted 87.32 (83.77; 90.94) μm. The percentage of EAT adipocytes with the size exceeding 100 μm indicated the degree of adipocytes hypertrophy; in our sample, this value was equal to 14.6 (8.9; 28.2)%. 

We divided all the study patients based on the median of the EAT adipocytes’ size into two groups: in patients from group 1, the mean size of EAT adipocytes was equal to or lower than 87.32 μm; patients from group 2 had the mean EAT adipocytes’ size above 87.32 μm. Medians of EAT adipocytes’ size differed by 7.16 μm (*p* < 0.001), while the median of the hypertrophic adipocytes’ percentage in group 2 exceeded the median in group 1 by 3 times (Table 2). 

Patients from group 2 had higher values of body mass index, waist and hip circumference (Table 2) as well as an elevated levels of fasting C-peptide and triglycerides in blood serum (Table 3). 

Patients with EAT adipocytes’ hypertrophy also had more severe atherosclerosis according to Gensini Score and frequency of multivessel CAD; however, the latter did not reach the level of statistical significance (Table 2). The frequency of patients with atrial fibrillation was comparable and negligible in both groups of patients (Table 2). 

The level of glycemia tended to increase in patients with more pronounced EAT adipocytes hypertrophy, even though we did not reveal any statistically significant changes of this parameter (Table 3). 

### 3.2. Biomarkers of Low-Grade Inflammation

Subjects with EAT adipocytes > 87.32 μm had higher concentrations of hsCRP, sPLA2 and TNF-α in blood serum, with IL-1β and LpPLA2 also tending to increase but not reaching the level of statistical significance (Figure 2).

### 3.3. Components of Extracellular Matrix Remodeling

The concentration of CTX-I was decreased in patients from group 2 compared to group 1, with both concentrations of MMPs and TIMP-1 being comparable between the groups of patients with different EAT adipocyte size (Figure 3).

### 3.4. Associations between EAT Adipocytes Size, Anthropometric, Biochemical Parameters and Low-Grade Inflammation

The size and percentages of hypertrophic adipocytes (>100 μm) directly correlated with BMI, waist and hip circumference (Table 4), while no correlation was observed with EAT thickness. At the same time, the percentage of small EAT adipocytes (<50 μm) showed an inverse correlation with EAT thickness (Table 4).

The size of EAT adipocyte and percentage of adipocytes > 100 μm directly correlated with fasting glycemia, fasting C-peptide, concentration of triglycerides, triglycerides/C-HDL ratio and apoB concentration (Table 5).

We observed direct correlations of EAT size and percentage of large EAT adipocytes with concentrations of IL-1β, TNF-α and LpPLA2, while percentage of small adipocytes was directly related to concentrations of IL-6, and inversely to concentrations of TNF-α (Table 6). Circulating CTX-I appeared to be the only biomarker of ECM remodeling, being inversely related to EAT size and percentage of large EAT adipocytes, with neither MMPs, nor TIMP-1 showing any correlations (Table 6).

The construction of a multiple linear regression model was impossible due to the lack of normality of the residuals of this model during the preliminary analysis of statistical data.

Multiple logistic regression allowed us to reveal the significant predictors of EAT adipocytes hypertrophy and classify patients into groups with its different intensity (Table 7).

The coefficient Nagelkerke of pseudo-determination of the constructed model was R_N_^2^ = 0.43. The significance level of the model was *p* = 0.0012. TNF-α had OR 2.615, 95% CI OR (1.025; 6.671); sPLA2 had OR 1.219, 95% CI OR (0.953; 1.561). The prognostic characteristics of the model were as followings: accuracy score 78%, sensitivity 71%, specificity 85%, AUC = 0.82. The ROC curve is represented in Figure 4. The cut-off level of the probability of EAT adipocytes hypertrophy constituted 0.56.

Even though BMI and waist circumference significantly differed between groups with various degree of EAT adipocytes hypertrophy (Table 2), these variables were not included in the model due to their statistically significant correlation with CTX-I (for BMI r_s_ = −0.545; *p* < 0.001; for waist circumference r_s_ = −0.584; *p* < 0.001; BMI significantly correlated with waist circumference: r_s_ = 0.793; *p* < 0.001).

## 4. Discussion

In our study, we demonstrated that increases in TNF-α, sPLA2 and a decrease in CTX-I represent the predictors of EAT adipocytes’ hypertrophy, and change of their concentration is associated not only with increment of EAT adipocytes’ size (above 87.32 μm), but also with a percentage of enlarged fat cells (above 14.6%). 

There are data indicating an association between EAT thickness with parameters of obesity and the presence of metabolic syndrome in patients without CAD [24,25]. In our study, the increase in EAT adipocytes correlated both with BMI and waist circumference but not with EAT thickness. There is some discrepancy of our data with research recently conducted by Aitken-Buck et al. (2019) [26], who did not reveal any association of EAT size neither with BMI nor with EAT thickness, pointing to the primacy of hyperplasia being involved in EAT total enlargement. One of the possible explanations of this data disparity may be the different methodological approach to the measuring of adipocyte size employed by the team of authors and in our work. Another possible factor that could have influenced the above-mentioned bias is samples variability. Our group of patients was approximately 10 years younger, and it included more women and mainly overweight and obese patients. As is known, age and gender represent important characteristics affecting adipose tissue mass and properties [1]. 

The question remains which pool of fat cells contributes to the EAT thickness increase, as in our study, the percentage of small adipocytes was inversely related to it. Earlier, it has been reported that metabolic impairments are associated with increment of small (<50 μm) dysfunctional fat cells’ numbers, which fail to expand appropriately and rather favor the development and sustainment of insulin resistance [27]. We cannot exclude that these are the medium-sized cells (>50 μm and <100 μm) that promote EAT thickening. Alternatively, it may be the stromal–vascular fraction and development of the fibrotic component that is involved in the growth of EAT thickness, as a lipid-enriched diet led to an upregulation of genes ECM remodeling in subcutaneous adipose without the recruitment of macrophages or other inflammatory cells [28]. At the same time, the fibrosis of adipose tissue is associated with the decrease in its plasticity and is inversely related to the size of adipocyte [29]. However, data for EAT depot are absent. 

The decrease in CTX-I associated with an enlargement of EAT adipocytes in our study is indicative for the impairments in the synthesis and breakdown of collagen type I. It is of great importance, as collagen type I, being the most prominent component of ECM, has been confirmed to influence adipocytogenesis in adipose-derived stem cells [30,31]. An in vitro culture of preadipocytes with collagen I promoted lypogenesis, an upregulation of the collagen receptors and a decrease in adiponectin [30]. On the other hand, collagen I suppressed adipogenesis through interaction with yes-associated protein (YAP) and the suppression of autophagy in 3T3-L1 preadipocyte cells [31]. A decrease in the concentration of its turnover products may point to the decline of the adipose tissue’s potential to enlarge via hyperplasia. It has to be kept in mind though that collagen I constitutes the main component of the structural framework mainly in subcutaneous adipose tissue [32], and it also represents the marker of bone turnover [33] as well as carotid plaque progression [34] and aorta calcification [35]. It has been demonstrated that postmenopausal women with increased VAT volume had lower levels of CTX-I in circulation [33]. The pathophysiological basis of this interconnection is not yet properly understood, but decreased bone turnover was associated with augmented insulin resistance and compromised glucose tolerance, which may impact the accumulation of VAT and fat cells’ hypertrophy [36]. The implication of EAT in this process also has not been studied and requires further investigation. Since ECM remodeling impairments are involved both in the progression of atherosclerosis [34,35] and development of EAT adipocytes hypertrophy, their interconnection deserves future in-depth research as well.

We did not reveal changes either in MMPs or in TIMP-1 concentrations depending on the fat cells morphology. However, it does not exclude their involvement in the modulation of adipocyte’s size and composition of EAT, as we measured only circulating levels of these agents. 

Experimental studies have revealed a close interconnection between adipocytes size, ECM remodeling and inflammation. However, relationships between collagen degradation and inflammation in metabolically unhealthy settings are not straightforward, as contradictory results have been obtained in different works. Mice with collagen IV knock-out displayed an increase in adipocytes size in epididymal and mesenteric fat tissues, which was associated with a decreased accumulation of macrophages [17]. In the study of postmenopausal women with H-type hypertension and osteoporosis, CTX-I was directly related to both TNF-α and IL-6 [37]. TNF-α-stimulated macrophages differentiated into foam cells to destroy collagen, which led to an increased production of CTX-I ex vivo [38]. At the same time, Djafari et al. (2021) demonstrated inverse relationships between urine CTX-I and body adiposity index, while the percentage of body fat was directly associated with serum hsCRP concentration [39].

We revealed that an increase in circulating inflammatory biomarkers was associated with an enlargement of EAT adipocytes. In particular, concentrations of hsCRP and TNF-α were higher in patients with elevated mean adipocyte size, while concentrations of IL-1β and TNF-α correlated with the percentage of hypertrophic EAT fat cells. The close link between TNF-α and EAT adipocytes hypertrophy was even more corroborated by the results of the multiple logistic analysis. 

EAT is currently regarded as an important endocrine organ with the potential to regulate systemic inflammation, providing the link between obesity and inflammatory response [40,41]. 

Multiple studies witnessed a correlation between EAT thickness and hsCRP concentration [24,42]; however, Carbone et al. (2019) demonstrated that this association is observed only in women [1].

TNF-α demonstrated a direct link with metabolic disturbances in multiple studies. Patients with diabetes mellitus type 2 presented an increase in plasmocytoid dendritic cells in adipose tissue associated with elevated circulating levels of TNF-α [10]. In addition, TNF-α may represent a metabolite, providing cross-talk between inflammatory adipose tissue and high cardiometabolic risk, as patients with rheumatoid arthritis responding to anti-TNF-α therapy had a lower frequency of myocardial infarction compared to non-responders [43]. Both adipocytes themselves and cells of stromal vascular fraction are capable of producing TNF-α, with inflammatory macrophages being the main source [44,45]. Kitagawa et al. (2018) demonstrated that TNF-α produced in fat performs primarily local or paracrine actions, as it did not correlate with EAT thickness and CRP measurements [11]. However, the size of adipocytes was not determined in this study; hence, one cannot exclude that the elevated levels of TNF-α observed in our research were of adipose origin. Effects of TNF-α on adipose tissue are primarily mediated through TNF-α receptor 1 (TNFR1) and include an increase in insulin resistance, induction of apoptosis and activation of pro-inflammatory signaling pathways, which in turn favors metabolic dysregulation [7,44]. In addition, TNF-α appeared to inhibit adipogenesis through the prevention of peroxisome proliferator-activated receptor gamma (PPARγ) and CCAAT-enhancer-binding protein alpha (C/EBPα) induction [46], which ultimately may be associated with the predominantly hypertrophic type of fat tissue expansion.

The sPLA2-IIA RNA was the most expressed gene in EAT compared to SAT according to the results of microarray gene expression profiling, with its expression greatly increasing in patients with CAD [9]. PLA2-IIA catalyzes the formation of inflammatory lipid mediators through the hydrolysis of membrane glycerophospholipids in a calcium-dependent manner. Macrophages are the most likely source of sPLA2 production [9], and inflammatory cytokines promote its synthesis, at least in human hepatoma cells [47]. The sPLA2-IIA in return may increase the local inflammation in EAT as it promotes the biosynthesis of PGE2, PGI2, MCP-1, IL-6, leptin and adiponectin in preadipocytes [12]. Since sPLA2 is also involved in membrane remodeling, one can expect its direct involvement in the regulation of adipocytes’ size. Indeed, mice with knock-out of PLA2G2E demonstrated not only a reduction in volumes of total, subcutaneous and visceral fat but also presented with a decrease in adipocyte size in perigonadal white adipose tissue [48]. 

Hypoxia may represent the driving force both of inflammatory impairments and failure of appropriate extracellular matrix degradation, which is associated with adipocytes’ hypertrophy. The enlargement of fat cells beyond the size necessary for the successful oxygen diffusion from capillaries leads to local hypoxia in adipose tissue [7,49]. Hypoxia leads to the activation of HIF-1α, which activates the synthesis of the components of ECM, affects collagen synthesis and stabilization, and promotes the development of fibrosis and adipose tissue dysfunction [50]. Even more so, hypoxia leads to the damage of adipocytes, which is followed by the accumulation of immune cells and increased production of inflammatory cytokines, inducing low-grade inflammation [8].

Neither biochemical parameter added to the prognostic significance for increased EAT adipocytes hypertrophy when included in the model of logistic regression together with TNF-α, sPLA2 and CTX-I. However, the size of adipocyte and percentage of hypertrophic adipocytes correlated with elevated glycaemia, hypertriglyceridemia, triglycerides/C-HDL ratio and the level of Apo-B, indicating the interconnection between the compromised metabolism and development of the pathological EAT.

Determination of fat cell size allows researchers to predict numerous obesity-related complications [51]. Even though we are not the first to demonstrate an association between the increased size of adipocyte and inflammatory response, the majority of data being previously published is based on the results obtained in SAT or omental VAT [15,52]. At the same time, the size of EAT and SAT adipocytes differs significantly, with EAT containing smaller cells due to the extended areas of immature preadipocytes and regions of inflammation-induced lipolysis [53]. This circumstance emphasizes the necessity of distinct studies to be performed in EAT to justify the extrapolation of the results obtained for SAT and VAT of other depots. Our constructed model of logistics regression provides possibilities to predict the presence of EAT adipocytes’ hypertrophy using a non-invasive approach based on the evaluation of inflammatory and ECM remodeling biomarkers in serum. The fact that EAT thickness evaluated by echocardiography did not correlate with the increased EAT cell size underscores the significance of our results even further. Moreover, it is primarily the size of adipocytes and not just total obesity that is interconnected with the inflammatory changes, as treatment with angiotensin-receptor antagonist led to the reduction in fat cell size but not BMI, and it was associated with change of inflammatory genes expression [52]. In other studies, it had been demonstrated that CAD exacerbated EAT adipocytes’ hypertrophy together with the degree of EAT meta-inflammation, which in turn may aggravate plaques progression and instability [53,54]. Indeed, patients of a group with enlarged EAT adipocytes in our work also were characterized by the higher levels of Gensini Score, which represents a cumulative index of atherosclerosis severity. In our previous work we also demonstrated that Gensini Score correlates both with the size of EAT adipocytes and the number of hypertrophied EAT fat cells [55].

Our research warrants further investigation of the association between the size of adipocytes, biomarkers of inflammation and ECM remodeling, as it may lead to new approaches of follow-up in patients with CAD. EAT has proven to be a distinct fat depot with unique properties. It possesses beige-like features, including the expression of uncoupling protein 1 (UCP-1) both at the mRNA and protein levels [56]. Adipocytes isolated from SAT demonstrated a high degree of plasticity and ability to differentiate into cells of other lineages, including myofibroblasts and fibroblasts [57]. Hence, the question remains as to whether the therapeutic potential revealed for SAT or beige adipocytes, such as their use in regenerative medicine [57] or inducible uncoupling conversion of energy-rich substrate molecules into ATP [58], will be applicable for EAT as well. In addition, various therapeutic approaches were proposed to modify the function of adipose tissue, such as glucagon-like peptide-1 receptor agonists, PPAR-γ agonists, selective cannabinoid receptor antagonists, β3-adrenoceptor agonists, etc. [59,60,61,62]. It is a prerequisite task to identify the action of these substances on EAT hypertrophy and the immunometabolic components associated with it. 

New simple inflammatory biomarkers, based on the results of the complete blood test with differential data, has recently been proposed, such as systemic inflammatory response index (SIRI), the aggregate index of systemic inflammation (AISI), and the systemic inflammatory index (SII) [63]. Their juxtaposition with the degree of EAT adipocytes hypertrophy, metabolic impairments and the severity of atherosclerosis will allow us to translate the obtained results into clinical settings. In addition, the implication of gender on EAT morphofunctional properties, meta-inflammation and ECM remodeling is of great importance in future larger scale studies, as multiple works have demonstrated an importance of sex in the stratification of dysmetabolic obese patients and fat distribution [1].

One of the limitations of our study is its relatively small size that did not allow us to further analyze the gender impact on the interconnection between the adipocyte size, metabolism and inflammation, which undoubtedly exists. In addition, the cross-sectional design of our study did not permit us to reveal implications of the crosstalk between adipocytes’ size and systemic inflammatory changes on the prognosis in patients.

## 5. Conclusions

We demonstrated for the first time that chronic low-grade inflammation and ECM remodeling are closely associated with the development of hypertrophy of epicardial adipose tissue adipocytes, with concentrations of TNF-α, sPLA2 and CTX-I in peripheral blood being the key predictors, describing the variability of epicardial adipocytes’ size.

## Figures and Tables

**Figure 1 biomedicines-11-00241-f001:**
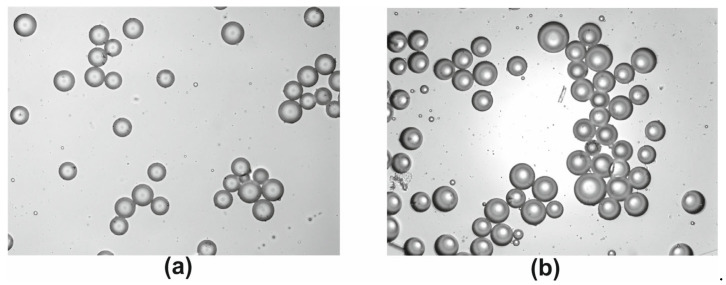
The representative images of EAT adipocytes of patients from groups 1 and 2. (**a**) Patient from the group of patients with non-hypertrophic EAT adipocytes, the median adipocyte diameter of this patient is 85.5 μm; (**b**) Patient from the group of patients with hypertrophied EAT adipocytes, the median adipocyte diameter of this patient is 90.45 μm; Light microscopy without staining, magnification × 200; EAT, epicardial adipose tissue.

**Figure 2 biomedicines-11-00241-f002:**
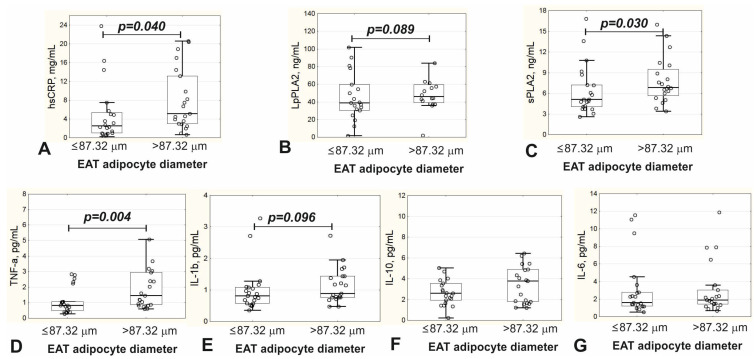
Serum concentrations of inflammatory biomarkers in patients with EAT adipocyte diameter ≤87.32 μm and >87.32 μm. (**A**) Concentration of hsCRP; (**B**) concentration of LpPLA2; (**C**) concentration of sPLA2; (**D**) concentration of TNF-α; (**E**) concentration of IL-1β; (**F**) concentration of IL-10; (**G**) concentration of IL-6; EAT, epicardial adipose tissue; hsCRP, high-sensitive C-reactive protein; LpPLA2, lipoprotein-associated phospholipase A2; sPLA2, secreted PLA2; TNF-α, tumor necrosis factor alpha; IL, interleukin; *p*, the level of statistical significance of the differences between patients with EAT adipocyte diameter ≤ 87.32 μm and patients with EAT adipocyte diameter > 87.32 μm.

**Figure 3 biomedicines-11-00241-f003:**
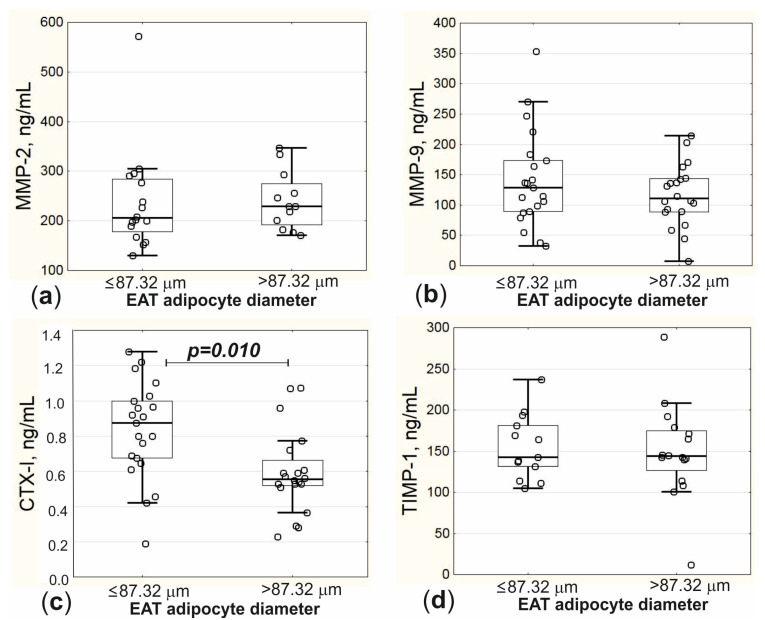
Serum concentrations of components of extracellular matrix remodeling in patients with EAT adipocyte diameter ≤ 87.32 μm and > 87.32 μm. (**a**) Concentration of MMP-2; (**b**) concentration of MMP-9; (**c**) concentration of CTX-I; (**d**) concentration of TIMP-1; EAT, epicardial adipose tissue; MMP, matrix metalloproteinase; CTX-I, C-terminal cross-linking telopeptide of type I collagen; TIMP-1, tissue inhibitor of matrix metalloproteinases type 1; *p*, the level of statistical significance of the differences between patients with EAT adipocyte diameter ≤ 87.32 μm and patients with EAT adipocyte diameter > 87.32 μm.

**Figure 4 biomedicines-11-00241-f004:**
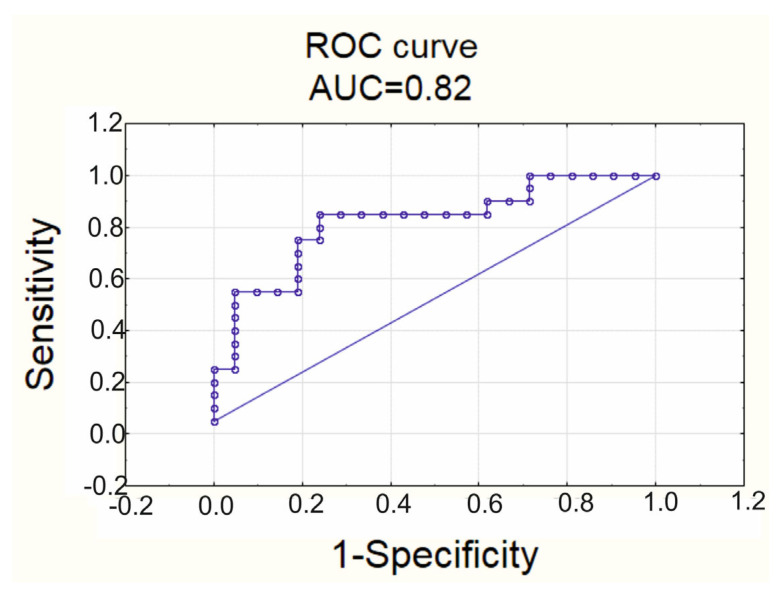
ROC curve of the multiple logistics regression for classification of patients into groups 1 and 2 with different hypertrophy of EAT adipocytes.

**Table 1 biomedicines-11-00241-t001:** Baseline characteristics of patients.

Parameters	
Gender (m/f)	28/14
Age, years	59 (56; 66)
History of myocardial infarction, *n* (%)	23 (54.8)
History of atrial fibrillation, *n* (%)	3 (7.1%)
Arterial hypertension, *n* (%)	42 (100)
Diabetes mellitus, *n* (%)	12 (28.6)
Gensini Score, points	78.0 (42.5; 121.0)
Multivessel CAD, *n* (%)	25 (59.5)
Ejection fraction, %	62.0 (47.0; 66.0)
Duration of arterial hypertension, years	20 (15; 23)
Duration of coronary artery disease, years	2 (2; 10)
Systolic blood pressure, mm Hg	130 (123; 140)
Diastolic blood pressure, mm Hg	80 (70; 85)
Smoking, *n* (%)	17 (40,5)
Body Mass Index, kg/m^2^	30.7 (28.1; 33.3)
Obesity, *n* (%)	25 (59.5)
Waist circumference, cm	106.4 (100; 116)
EAT thickness, mm	5.23 (4.37; 6.30)
Metformin, *n* (%)	10 (23.8)
RAAS inhibitors, *n* (%)	36 (85.7)
Calcium channels antagonists, *n* (%)	26 (61.9)
Beta-blockers, *n* (%)	38 (90.5)
Diuretics, *n* (%)	16 (38.1)
Statins, *n* (%)	41 (97.6)

Note: data are represented as median and interquartile interval Me (Q1; Q3); EAT, epicardial adipose tissue; RAAS, renin–angiotensin–aldosterone system.

**Table 2 biomedicines-11-00241-t002:** Anthropometric characteristics of obesity and morphometric parameters EAT adipocytes depending on the mean size of adipocytes.

Parameters	Group 1 Diameter of EAT Adipocytes ≤ 87.32 μm (*n* = 21)	Group 2 Diameter of EAT Adipocytes > 87.32 μm (*n* = 21)	*p*
Men/women, *n*%	13 (61.9%)/8 (38.1%)	15 (71.4%)/6 (28.6%)	0.530
Body Mass Index, kg/m^2^	29.2 (27.0; 31.2)	32.8 (29.9; 35.4)	0.036
Waist circumference, cm	102 (76; 128)	112 (99; 122)	0.010
Hip circumference, cm	104 (101; 108)	110 (105; 117)	0.037
EAT thickness, mm	5.05 (4.37; 6.15)	5.45 (4.35; 6.60)	0.690
EAT adipocyte size, µm	83.79 (79.27; 86.03)	90.95 (89.09; 95.13)	<0.001
%EAT adipocytes > 100 µm	8.88 (5.91; 10.82)	28.22 (18.24; 34.75)	<0.001
%EAT adipocytes < 50 µm	1.58 (1.04; 2.89)	1.2 (0.27; 3.85)	0.630
History of myocardial infarction, *n* (%)	11 (52.4)	12 (57.1)	0.999
History of atrial fibrillation, *n* (%)	1 (4.8)	2 (9.5)	0.999
Gensini Score, points	70.0 (28.0; 100.0)	107.0 (71.0; 144.0)	0.008
Multivessel CAD, *n* (%)	9 (42.9)	16 (76.2)	0.058
Ejection fraction, %	64.0 (47.0; 69.5)	61.0 (48.0; 64.0)	0.346

Notes: Data are represented as median and interquartile interval Me (Q1; Q3); *p*, the level of statistical significance of the differences of the parameter in groups 1 and 2 according to the Mann–Whitney rank test.

**Table 3 biomedicines-11-00241-t003:** Parameters of glucose/insulin metabolism and lipid transport blood function depending on the mean size of adipocytes.

Parameters	Total (*n* = 42)	Group 1 Diameter of EAT Adipocytes ≤ 87.32 μm (*n* = 21)	Group 2 Diameter of EAT Adipocytes > 87.32 μm (*n* = 21)	*p*
Fasting glucose, mM	5.89 (5.30; 6.60)	5.6 (5.1; 6.04)	6.21 (5.70; 6.79)	0.078
Fasting insulin, µIU/mL	5.1 (2.64; 7.84)	6.02 (2.64; 8.65)	4.65 (2.66; 5.4)	0.300
Triglycerides, mM	1.41 (1.08; 1.92)	1.2 (0.9; 1.37)	1.7 (1.41; 2.16)	0.007
Fasting C-peptide, ng/mL	2.3 (1.97; 2.98)	2.15 (1.73; 2.75)	2.78 (2.18; 3.1)	0.030
HDL cholesterol, mM	1.03 (0.88; 1.18)	1.04 (0.88; 1.18)	0.99 (0.86; 1.16)	0.400
LDL cholesterol, mM	2.04 (1.72; 2.42)	2.0 (1.75; 2.4)	2.21 (1.62; 2.56)	0.082
Apolipoprotein B, mg/dL	105.28 (8.5; 131.29)	90.16 (67.28; 112.34)	112.34 (91.82; 136.82)	0.080
Apolipoprotein A1, mg/dL	146.20 (124.57; 168.42)	140.12 (121.07; 167.56)	148.06 (134.13; 168.42)	0.490
apoB/apoA1	0.69 (0.49; 0.83)	0.61 (0.45; 0.76)	0.73 (0.55; 0.92)	0.230

Notes: Data are represented as median and interquartile interval Me (Q1; Q3); *p*, the level of statistical significance of the differences of the parameter in groups 1 and 2 according to the Mann–Whitney rank test.

**Table 4 biomedicines-11-00241-t004:** Correlations of EAT adipocyte size, percentages of large and small adipocytes with anthropometric parameters of obesity and EAT thickness.

	EAT Adipocyte Size		Percentage of EAT Adipocytes > 100 μm		Percentage of EAT Adipocytes < 50 μm	
Parameters	r_s_	*p*	r_s_	*p*	r_s_	*p*
BMI	0.46	**0.002**	0.49	**0.001**	0.07	0.650
Waist circumference	0.55	**<0.001**	0.54	**<0.001**	0.10	0.530
Hip circumference	0.39	**0.010**	0.43	**0.004**	0.13	0.410
WHR	0.37	**0.017**	0.33	**0.030**	−0.13	0.400
EAT thickness	0.18	0.280	0.13	0.430	−0.32	**0.048**

Notes: BMI, body mass index; WHR, waist-to-hip ratio; bold values indicate significant correlations.

**Table 5 biomedicines-11-00241-t005:** Correlations of EAT adipocyte size, percentages of large adipocytes with metabolic parameters.

	EAT Adipocyte Size		Percentage of EAT Adipocytes > 100 μm	
Parameters	r_s_	*p*	r_s_	*p*
Fasting glycemia	0.40	**0.009**	0.44	**0.003**
Fasting C-peptide	−0.16	0.290	−0.03	0.850
Triglycerides	0.49	**0.001**	0.48	**0.001**
Triglycerides/HDL cholesterol	0.37	**0.017**	0.33	**0.030**
LDL cholesterol	0.09	0.590	0.15	0.330
ApoB	0.36	**0.034**	0.34	**0.047**
ApoA1	0.18	0.302	0.05	0.793

Notes: HDL, high-density lipoprotein; LDL, low-density lipoprotein; bold values indicate significant correlations.

**Table 6 biomedicines-11-00241-t006:** Correlations of EAT adipocyte size, percentages of large and small adipocytes with biomarkers of inflammation and extracellular matrix remodeling.

	EAT Adipocyte Size		Percentage of EAT Adipocytes > 100 μm		Percentage of EAT Adipocytes < 50 μm	
Parameters	r_s_	*p*	r_s_	*p*	r_s_	*p*
IL-1β	0.34	**0.026**	0.31	**0.048**	−0.29	0.060
IL-6	−0.12	0.450	−0.001	0.990	0.40	**0.009**
IL-10	0.14	0.370	0.22	0.170	0.03	0.870
TNF-α	0.49	**0.001**	0.41	**0.006**	−0.33	**0.030**
LpPLA2	0.34	**0.040**	0.44	**0.008**	0.08	0.630
sPLA2	0.23	0.150	0.25	0.110	−0.17	0.300
hsCRP	0.15	0.340	0.22	0.170	0.06	0.730
CTX-I	−0.43	**0.005**	−0.41	**0.007**	−0.13	0.42

Notes: IL, interleukin; TNF-α, tumor necrosis factor alpha; LpPLA2, lipoprotein-associated phospholipase A2; sPLA2, secreted phospholipase A2; hsCRP, high sensitive C-reactive protein; CTX-I, C-terminal cross-linking telopeptide of type I collagen; bold values indicate significant correlations.

**Table 7 biomedicines-11-00241-t007:** Estimates of the multiple logistics regression model and their significance level.

Parameters	Estimate	*p*
Intercept	0.438	0.751
TNF-α	0.961	0.044
sPLA2	0.198	0.116
CTX-I	−4.405	0.009

Notes: TNF-α, tumor necrosis factor alpha; sPLA2, secreted phospholipase A2; CTX-I, C-terminal cross-linking telopeptide of type I collagen.

## Data Availability

The raw data are available from the corresponding author upon the reasonable request.

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
