# Peer review of "Association of Epicardial Adipose Tissue Adipocytes Hypertrophy with Biomarkers of Low-Grade Inflammation and Extracellular Matrix Remodeling in Patients with Coronary Artery Disease"

_biomedicines, 2023, doi:10.3390/biomedicines11020241_

Round 1
Reviewer 1 Report
The authors have conducted an interestingly and timely research about the connection between biomarkers of low-grade inflammation such as TNF, sPLA2 and 439 CTX-I, matrix remodeling and epicardial adipose tissue adipocytes hypertrophy.
Minor recommendation:
|2.1. . Patients, and 2.2. . Echocardiography” needs corrections;
In tables, I suggest to give in legend an explanation for the bold values;
Row 274, “TNF- “ needs “α” to be added;
Row 306, “In in vitro study”,, repetition;
Row 351, “at al. (2019)”, needs corrections into et. al.
Row 369, the meaning for PPARγ and C/EBP should be included;
Author Response
Dear Reviewer!
Thank you for your efforts that you put working on our article! It had certainly led to its improvement. We corrected our manuscript according to your suggestions.
- |2.1. . Patients, and 2. . Echocardiography” needs corrections
- We introduced the correction
- In tables, I suggest to give in legend an explanation for the bold values;
- We added explanation under the Tables 4, 5, 6
- Row 274, “TNF- “ needs “α” to be added;
- We have added the symbol “α”
- Row 306, “In in vitro study”,, repetition;
- We have corrected the sentence
- Row 351, “at al. (2019)”, needs corrections into et. al.
- We introduced the correction
- Row 369, the meaning for PPARγ and C/EBP should be included
- We have deciphered the above mentioned abbreviations.
Best regards,
The team of authors
Reviewer 2 Report
This is an interesting study about association of epicardial adipose tissue hypertrophy and markers of low-grade inflammation and MMP\TIMP system. Authors collected epicardial adipose tissue specimens in 42 patients during GABG surgery that made this study unique. However, several major issues have to be corrected or mentioned.
Abstract
Represented the body of the paper.
Background
In the Introduction section the authors reflected the current knowledge about epicardial adipose tissue.
Materials and methods
Baseline patients characteristic have to include data about LV function, CAD severity and atrial fibrillation. There is an increasing amount of publications on impact of epicardial adipose tissue on atrial fibrillation incidence etc. Those clinical data have to be mentioned.
Statistics is essential
Results
The major question to the paper is about the problem that was investigated by authors.
Two associations were found. Number one is obesity and epicardial cell hypertrophy. Patients with EAT dimensions above median differ significantly in BMI and WC. Second is correlation between EAT hypertrophy and pro-inflammatory state. Does it mean that the results of the paper only supported well-known thesis about low-grate inflammation in patients with obesity? Thus, novelty of the study seems to be limited. It can be recommended to include BMI or WC to the model etc.
Figures are good.
Discussion
In that part authors compare own results with published. Have to be partly corrected after changings in the Results section.
Author Response
Dear Reviewer!
Thank you for your efforts that you put working on our article! It had certainly led to its improvement. We corrected our manuscript according to your suggestions:
- Materials and methods
Baseline patients characteristic have to include data about LV function, CAD severity and atrial fibrillation. There is an increasing amount of publications on impact of epicardial adipose tissue on atrial fibrillation incidence etc. Those clinical data have to be mentioned. Statistics is essential.
We have retrieved additional data from patients’ history and inserted them both in Tables 1 and 2. Table 1 had already reflected the severity of atherosclerosis, represented by Gensini Score, but we also added data on the frequency of multivessel CAD in patients. Gensini Score was higher in the group of patients with EAT adipocytes’ hypertrophy. We added a brief discussion of this finding and provided a link to our previous work, where interconnection between EAT cell size and the severity of atherosclerosis was studied in more details (Link 55 in the revised version of the manuscript). The frequency of atrial fibrillation was negligible (was present only in 3 patients), which we described in 3.1.
- Results
The major question to the paper is about the problem that was investigated by authors. Two associations were found. Number one is obesity and epicardial cell hypertrophy. Patients with EAT dimensions above median differ significantly in BMI and WC. Second is correlation between EAT hypertrophy and pro-inflammatory state. Does it mean that the results of the paper only supported well-known thesis about low-grate inflammation in patients with obesity? Thus, novelty of the study seems to be limited. It can be recommended to include BMI or WC to the model etc.
We have consulted with our specialist on medical statistics, and she insisted not to include BMI and WC into the present model, as they possess a high degree of collinearity with each other and CTX-I. We have described this fact in the end of the Results section.
We tried to stress an importance of the interconnection between the studied parameters and the degree of EAT adipocytes’ hypertrophy, as this adipose tissue depot remains not so well studied as subcutaneous and omental fat tissues. The fragment of discussion is added, as well as additional links – 51-53. The creation of the model of logistic regression adds a potential practical significance to our study, which can be elaborated in the future. We hope that it added emphasize to the novelty of the study.
- Discussion
In that part authors compare own results with published. Have to be partly corrected after changings in the Results section.
We did introduce some corrections, as it is described above.
- We had also asked a native English speaker to read through our manuscript, who had suggested some minor grammar and style corrections, which were introduced accordingly.
Best regards,
The team of authors
Round 2
Reviewer 2 Report
The paper can be recommended for the Journal